# STING Orchestrates EV-D68 Replication and Immunometabolism within Viral-Induced Replication Organelles

**DOI:** 10.3390/v16101541

**Published:** 2024-09-29

**Authors:** Kathy Triantafilou, Barbara Szomolay, Mark William Shepherd, Joshi Ramanjulu, Martha Triantafilou

**Affiliations:** 1Division of Infection and Immunity, School of Medicine, University Hospital of Wales, Cardiff University, Heath Park, Cardiff CF14 4XN, UK; triantafilouk@cardiff.ac.uk (K.T.); szomolayb@cardiff.ac.uk (B.S.); shepherdm6@cardiff.ac.uk (M.W.S.); 2Immunology Research Unit, GlaxoSmithKline, 1250 South Collegeville Road, Collegeville, PA 19426, USA

**Keywords:** stimulator of interferon genes (STING), Enterovirus D-68, replication organelles, SARS-CoV-2, glucose metabolism

## Abstract

Some respiratory viruses, such as Human Rhinovirus, SARS-CoV-2, and Enterovirus D-68 (EV-D68), share the feature of hijacking host lipids in order to generate specialised replication organelles (ROs) with unique lipid compositions to enable viral replication. We have recently uncovered a novel non-canonical function of the stimulator of interferon genes (STING) pathway, as a critical factor in the formation of ROs in response to HRV infection. The STING pathway is the main DNA virus sensing system of the innate immune system controlling the type I IFN machinery. Although it is well-characterised as part of the DNA sensor machinery, the STING function in RNA viral infections is largely unexplored. In the current study, we investigated whether other RO-forming RNA viruses, such as EV-D68 and SARS-CoV-2, can also utilise STING for their replication. Using genetic and pharmacological inhibition, we demonstrate that STING is hijacked by these viruses and is utilised as part of the viral replication machinery. STING also co-localises with glycolytic enzymes needed to fuel the energy for replication. The inhibition of STING leads to the modulation of glucose metabolism in EV-D68-infected cells, suggesting that it might also manipulate immunometabolism. Therefore, for RO-generating RNA viruses, STING seems to have non-canonical functions in membrane lipid re-modelling, and the formation of replication vesicles, as well as immunometabolism.

## 1. Introduction

All positive-sense RNA (+RNA) viruses, including hepatitis C virus (HCV) [1], dengue virus [2], Zika virus [3] and polioviruses [4], Human Rhinovirus (HRV), Enterovirus D68 (EV-D68), and Severe acute respiratory syndrome coronavirus 2 (SARS-CoV-2), like all coronaviruses, share the feature of establishing specialised membranous replication organelles (ROs) with unique lipid compositions to enable robust viral replication [5,6]. The ROs provide an optimal microenvironment for the synthesis of viral RNA by concentrating viral components (RNA and proteins) and recruiting host receptors and lipids required for viral RNA synthesis. In addition to creating a distinct subcellular microenvironment to facilitate replication, ROs protect viral RNAs against degradation by cellular RNases and detection by the host innate immune defences [7]. Although studies have shed light into the replication organelles that are induced by +RNA viruses, the link between these structures to host cellular compartments as well as the recruitment of host cell molecules within these are poorly understood. In our recent work, we have uncovered a novel non-canonical function of STING as a critical factor in the formation of HRV-induced ROs and an essential molecule for HRV replication and transmission [8].

The cyclic GMP-AMP synthase (cGAS)–stimulator of interferon genes (STING) pathway is the main DNA virus sensing system of the innate immune system controlling the type I IFN machinery [9,10,11,12], as well as inducing autophagy for the clearance of cytosolic DNA through a TANK binding kinase (TBK1) independent mechanism [13]. Although it is well-characterised as part of the DNA sensing machinery, STING’s function in RNA viral infections is largely unexplored. Our work has revealed a novel non-canonical role for STING in RNA viral infections, where it is essential for RO formation and viral replication [8]. STING was shown to be essential for viral replication since it interacted with Phosphatidylinositol 4-phosphate (PI4P) and was trafficked to replication organelles (ROs) created by the virus to facilitate HRV replication and transmission via autophagy [8].

The question now is whether this is a general mechanism across RNA viruses—are STING and PI4P also important for other respiratory RNA viruses, such as coronaviruses (HCoVs) and EV-D68, which also generate ROs? Coronaviruses are associated with a number of infectious disease outbreaks in humans, including SARS in 2002–2003 and Middle East respiratory syndrome (MERS) in 2012 [14,15], as well as causing the COVID-19 global pandemic which has killed millions of people worldwide [16]. Together with coronaviruses, HRVs are responsible for the majority of respiratory tract infections in all age groups [17,18,19], as well as being a major cause of morbidity in infants, young children, and the elderly [19,20]. EV-D68 is an ssRNA enterovirus that is of emerging worldwide public-health concern, due to its severe respiratory illness, as well as the acute flaccid paralysis myelitis (AFM) it can cause in children [21]. The United States (US) experienced an unprecedented outbreak in 2014 of EV-D68-induced respiratory disease that was associated with the emergence of acute flaccid myelitis (AFM), a paralytic disease occurring mainly in children, that has a striking resemblance to poliomyelitis [22]. To date, there are no approved treatments for EV-D68.

The current study builds on our previous findings and explores the role of STING in other RO-mediated RNA viruses. Using the genetic as well as pharmacological inhibition of STING, we demonstrate that STING is essential for the replication of RO-generating viruses such as EV-D68 as well as SARS-CoV-2. The viruses seem to hijack STING into their replication organelles, impairing its canonical anti-viral function. Within the organelles, STING not only is essential for viral replication but also modulates the energy needed for replication. The inhibition of STING leads to the modulation of glucose metabolism in EV-D68-infected cells. Therefore, for RO-generating RNA viruses, STING seems to have non-canonical functions in membrane lipid re-modelling, and the formation of replication vesicles, as well as immunometabolism.

## 2. Materials and Methods

### 2.1. Analysis of Publicly Available Transcriptomic Data

Genome-wide transcriptional profiling of EV-D68-infected human rhabdomyosarcoma (RD) cells was performed at different time points [23], where total RNA from non-infected and EV-D68-infected RD cells was analysed using RNA-sequencing to quantify differentially expressed genes (DEGs) relative to the mock control group. Figure 1A,B and Figure 5A–C show relevant DEGs satisfying the threshold values |log2FC| > 0.3 & padj < 0.05 at 12 and 24 h post infection (hpi) for three pathways: CGAS/STING, PI4P/OBP, and glycolysis pathways. CGAS/STING and PI4P/OBP pathways were selected using Reactome, and the glycolysis pathway was selected using the combination of Reactome, KEGG, PANTHER, and MSigDB databases. Figures were generated with the EnhancedVolcano R package, and R-4.3.3.

### 2.2. Materials

Anti STING rabbit polyclonal antibody (PA5-23381) was purchased from Invitrogen (Carlsbad, CA, USA). Anti STING goat antibody (sc241046) was purchased from Santa Cruz (Dallas, TX, USA); J2 mAb (English & Scientific Consulting, Szirák, Hungary); PI4P IgM mAb (Echelon, Salt Lake City, UT, USA); Anti Calreticulin goat polyclonal (PA1-33045) (Thermofischer Scientific, Newport, UK) was used to stain Endoplasmic reticulum. ERGIC-53 monoclonal antibody (OTI1A8) from ENZO (Exeter, UK) as well as ERGIC-53 monoclonal antibody (E1031) from Sigma-Aldrich (Dorset, UK) were used. Alexa Fluor-488 or -546 labelled secondary antibodies against mouse or rabbit IgG or IgM (Thermofisher Scientific, Newport, UK) were used. Poly (I:C)/Lyovec as well as 2′3′-cGAMP were purchased from Invivogen (Toulouse, France). 2′3′-cGAMP (5 µg/mL) was added to the cells complexed with LyoVec (Invivogen) in order to aid internalisation. STING antagonist GSK′783 was provided by Dr Joshi Ramanjulu from GSK. STING antagonist H-151, cGAS inhibitor RU.521, and TBK-1 inhibitor BX795 were purchased from Invivogen (Toulouse, France).

### 2.3. Cell Culture/Viruses

Bronchial epithelial cells were cultured in RPMI medium containing 10% FBS, and 1% non-essential amino acids. Bronchial epithelial cells were seeded in Lab-Tek 8-well slides (80,000 cells/well) for two days. Once they were ~80% confluent, they were incubated with the different viruses (MOI:5) in 500 µL of serum-free medium (SFM) for 2 h at 37 °C. Following the stimulation, the supernatant was removed and the cells were washed ×2 with PBS, followed by fixation using 500 µL per well formalin for 15 min at RT.

Air–liquid interfaces (ALIs) of human bronchial epithelial cells (hBECs) from normal donors were purchased from Lonza and grown on transwells and cultured in Pneumacult media (Seeding density: 150,000 cells/well); age of cells used in experiments post air-lift was 14 days.

EV-D68 was provided by Professor Frank van Kuppeveld. Human Rhinovirus 1B (HRV1B) was purchased from Virapur (San Diego, CA, USA). Influenza A virus (IAV) H3N2 was purchased from ATCC (Teddington, UK). SARS-CoV-2 strain 2019-nCoV/Italy/INMI1 was used in this study. The virus was amplified in Vero E6 cells in high-glucose DMEM supplemented with 2% FBS, incubated at 37 °C in 5% CO_2_ during 2 to 4 days of infection. Virus titers were performed by the tissue culture infectious dose at 50% (TCID50/mL) and the virus stocks kept in −80 °C freezers. According to WHO guidelines, all procedures involving virus culture were performed in biosafety level 3 (BSL3) multiuser facility. SARS-CoV-2 infections were performed at MOI of 0.01 in all cells.

### 2.4. Confocal Microscopy

After fixation, the cells were washed ×2 with PBS and were permeabilised using 500 µL per well of PBS/0.02% BSA/0.02% Saponin and labelled with antibodies specific for PI4P, STING, and J2 mAb to detect viral dsRNA (1:500), as well as secondary antibodies conjugated to the appropriate fluorophore (1:500). Following the secondary antibody incubation, the cells were washed three times using PBS/0.02% BSA/0.02% NaN_3_. Finally, all liquid was removed, as well as the plastic inserts. The cells were mounted with Vectashield, covered with a coverslip and sealed using clear nail varnish.

Cells were imaged on a Carl Zeiss, Inc. (Jena, Germany) LSM710 ELYRA P1 confocal using a 1.4 NA 63× Zeiss objective. The images were analysed using LSM 2.5 image analysis software (Carl Zeiss, Inc.). No fluorescence was observed from an Alexa 488-labelled specimen using the 594 filters, nor was 594 fluorescence detected using the 488 filter sets.

The degree of co-localisation was quantified using Costes’ approach [23]. Coste’s approach, Pearson’s correlation coefficients, and *p*-values were calculated using MBF ImageJ with Just Another Co-localisation Plugin (JACoP) (http://macbiophotonics.ca/). Values greater than 0.5 are considered significant co-localisation.

The following gRNA sequences were designed by Feng Zhang’s laboratory at the Broad Institute to uniquely target the STING1 gene within the human genome.

### 2.5. CRISPR Knockout

The following gRNA sequences were designed by Feng Zhang’s laboratory at the Broad Institute to uniquely target the STING1 gene within the human genome. These gRNA sequences are for use with WT SpCas9, or as crRNA for use with WT SpCas9 protein, to introduce a DSB for genome editing. The following gRNA sequences were designed (GENscript) to uniquely target genes within the human genome. These gRNA sequences were used with CRISPR/Cas9 Knockout (KO) Plasmid, to introduce a DSB for genome editing:
MDA5gRNA Target sequence CGAATTCCCGAGTCCAACCA
gRNA Target sequence AGCGTTCTCAAACGATGGAGSTINGgRNA Target sequence GCGGGCCGACCGCATTTGGG
gRNA Target sequence GGTGCCTGATAACCTGAGTARIG-IgRNA Target sequence GGGTCTTCCGGATATAATCC
gRNA Target sequence TTGCAGGCTGCGTCGCTGCT

Bronchial epithelial cells were seeded at a density of 0.8–3.0 × 10^5^ cells/mL in 12-well plates. Then, 50 nM guide RNA and 2.5 µg plasmid DNA encoding Cas9 were transfected to cells via *Trans*IT-X2 (MIRUS LLC, Madison, NY, USA) following manufacturer’s recommendations.

## 3. Results

### 3.1. Involvement of STING in RNA Viruses That Utilise ROs

In order to test our hypothesis that STING is also important for other respiratory RNA viruses that utilise ROs, such as SARS-CoV-2 and EV-D68, we looked at transcriptomics from publicly available databases [23,24]. A bioinformatics analysis of the single-cell and bulk RNA-seq profiling of COVID-19 patients’ data revealed a significant upregulation of cGAS-STING signalling pathway genes (TMEM173, IRF3, NFKB1, CGAS, IFNAR1, and TBK1) in the lung (Appendix A). In addition, a bioinformatics analysis of the bulk RNA-sequencing profiling of cells infected with EV-D68 over multiple timepoints [23] was performed. Total RNA from non-infected and EV-D68-infected cells was analysed using RNA-sequencing to quantify differentially expressed genes (DEGs) relative to the mock control group. DEGs at 12 hpi (Figure 1A) and 24 hpi (Figure 1B) for the CGAS/STING and the PI4P/OSPB pathway as a control were analysed. At 12 hpi, six DEGs (TMEM173, NFKB2, RELA, IKBKG, IRF3, and NFKB1) involved in CGAS/STING were upregulated (Figure 1A), and four of these (TMEM173, NFKB2, RELA, and IRF3) also had a high expression at 24 hpi (Figure 1B). Out of the DEGs involved in PI4P/OSBP, OSBPL11 remained downregulated, whereas OSBPL2 and OSBBPL10, were upregulated, at both time points. In the middle stage of infection, OSBPL1A and PIP4K2G, and, in the late stage of infection, PIP3K2B, PI4KA, and PI4K2A, were differentially expressed, revealing a profound transcriptional dysregulation of host genes of the PI4P/OSBP pathway, thus suggesting the importance of STING as well as the PI4P pathway for RO-generating RNA viruses.

Since the bioinformatics analysis of transcriptomics suggested the involvement of STING in EV-D68 as well as SARS-CoV-2 infections, we proceeded to investigate the localisation of STING in response to these viruses. Our confocal data demonstrated that both EV-D68 and SARS-CoV-2 co-localise with STING in ROs (Figure 1C,D) in bronchial epithelial cells. In addition, we investigated the location of STING in response to EV-D68 infection in air–liquid interface (ALI) hBECs (Figure 1D). It was shown that, in response to EV-D68 infection, STING co-localised with PI4P and EV-D68 dsRNA in the ALI, suggesting that STING is an important molecule in RNA-virus induced ROs.

### 3.2. Non-Canonical Function of STING Is Essential for EV-D68 Replication

To investigate what the function of STING in EV-D68 infections is, *Tmem173* (hereby referred to as STING) knockouts were generated by CRISPR-Cas9 editing in both BEAS-2B and air–liquid interface primary human airway epithelial (ALI) cells. STING deficiency was confirmed using flow cytometry, where no expression of STING protein was detected in STING-knockout (KO) cells (Appendix A), as well as using a functional assay to stimulate the cells with 2′3′-cGAMP, which is a STING agonist (Appendix A). There was no IFN-β production in response to 2′3′-cGAMP in STING-KO cells. Once it was verified that STING-KO cells did not express STING and could not respond to its ligand, STING-deficient cells were infected with EV-D68 (Figure 2A) as well as other RNA respiratory viruses such as Human Rhinovirus 1B (HRV-1B) (Figure 2B) SARS-CoV-2 (Figure 2C) and Influenza A virus (IAV) H3N2 (Figure 2D). Viral replication was assessed using qPCR and confirmed that STING downregulation abrogated EV-D68 replication. As we have previously demonstrated, HRV replication (Figure 2B) was also abrogated as well as SARS-CoV-2 (Figure 2C), whereas IAV infection was not affected (Figure 2D).

In addition to genetic inhibition, we verified STING’s involvement in the replication of EV-D68 using pharmacological inhibition. Prior to using the inhibitors for our experiments, we proceeded to verify that the STING, cGAS, and TBK1 antagonists work as intended and inhibit STING-induced IFN-β production (Appendix A). STING inhibitors, GSK′783, as well as commercially available H-151, were able to inhibit EV-D68 replication (Figure 2E), as well as HRV (Figure 2F) and SARS-CoV-2 (Figure 2G), whereas they were unable to inhibit IAV replication (Figure 2H).

To determine whether the cGAS-STING pathway is involved in EV-D68 replication, we also tested inhibitors upstream and downstream of STING. Inhibitors for cGAS (RU.521), as well as TBK-1 (BX795), were used prior to infection. It was shown that neither the cGAS inhibitor nor the TBK-1 inhibitor were able to abrogate EV-D68 replication (Figure 2I), thus verifying that it is only STING that is involved in EV-D68 replication.

### 3.3. STING Does Not Contribute to Type-I Interferon Production in Response to EV-D68

The innate immune system is the front line of defence against viruses with the cGAS-STING pathway as the main anti-viral DNA sensing system controlling the type I IFN machinery. In contrast to its well-documented role in antiviral immune responses to DNA viruses, the STING function against RNA virus infections is largely unexplored. Although EV-D68 is an RNA virus, we investigated whether there was interferon-β (IFN-β) production in response to EV-D68 infection in bronchial epithelial cells. It was shown that bronchial epithelial cells produced IFN-β in response to 2′3′-cGAMP, but also EV-D68 infection (Figure 3A). GSK′783, as well as H-151, which are STING inhibitors, did not inhibit EV-D68-induced IFN-β production (Figure 3A). In addition, we tested inhibitors upstream and downstream of STING in the pathway. BX795, a TBK1 inhibitor, and RU.521, a cGAS inhibitor, did not inhibit the EV-D68-induced IFN-β production (Figure 3B), demonstrating cGAS-STING-independent IFN-β production in response to EV-D68 infection.

We have previously shown that IFN production upon HRV infection is RNA-helicase-dependent; therefore, it is possible that EV-D68 being in the same enteroviral family could also be triggering RNA-helicase-dependent IFN-β production. In order to investigate this, we knocked out MDA5 and RIG-I using CRISPR. Knockout was confirmed using flow cytometry to demonstrate that RIG-I/MDA5 protein expression was inhibited (Appendix A), as well as a functional assay using Poly(I:C)-induced IFN-β production, demonstrating that RIG-I/MDA5 KO cells could not respond to Poly(I:C) (Appendix A). Following the verification of knockout, it was shown that, whilst STING-deficient and wild-type cells were able to generate an IFN-response, MDA5/RIG-I KO cells were not (Figure 3C), suggesting that RLRs, and not STING, drive the IFN-response to EV-D68 infection. Interestingly, when bronchial epithelial cells were challenged with EV-D68 followed by 2′,3′-cGAMP, STING was unable to signal as expected with no IFN response produced (Figure 3D). These results collectively indicate that RLRs, and not STING, are responsible for downstream IFN production following EV-D68 infection in respiratory cells, and, more interestingly, that STING cannot canonically function following EV-D68 infection.

### 3.4. STING Resides in PI4P-Rich Viral Replication Organelles

It is well-established that, upon DNA-triggered activation, STING traffics from the ER to the ER–Golgi intermediate compartment (ERGIC), and then to the Golgi to trigger signalling via TBK1 and IRF3 [25]. In its resting state, STING localises to the ER membrane and is retained there by interacting with stromal interaction molecule 1 (STIM1), a Ca^2+^ sensor [26].

Emerging evidence reveals that STING location dictates function [27]. The transportation of STING to different organelles dictates immune-dependent functions (i.e., Type I interferon response), or immune-independent functions (i.e., the activation of autophagy [13], cell death [28], ER stress [29], lipid metabolism [30], or the formation of viral ROs [8]).

Our data suggested that EV-D68 interfered with the canonical STING signalling; therefore, we proceeded to investigate the intracellular location of STING in response to EV-D68 infection. We had already observed using confocal microscopy that STING co-localised with PI4P and EV-D68 dsRNA in replication organelles (Figure 1C). In order to verify its location following EV-D68, we also utilised fluorescence resonance energy transfer (FRET). FRET can occur over 1–10 nm distances, and effectively increases the resolution of light microscopy to the molecular level. The data confirmed that, in response to 2′3′-cGAMP, STING interacted with ERGIC and was not interacting with either STIM1 or PI4P (Figure 4A). Upon EV-D68 infection, it was shown that STING interacted with PI4P and not ERGIC or STIM1 (Figure 4A). The pharmacological inhibition of STING during EV-D68 infection, using STING antagonist GSK′783, demonstrated that STING did not associate with PI4P but instead remained in the ER and associated with STIM1 (Figure 4B), confirming that EV-D68 interferes with STING canonical trafficking upon infection. This aberrant trafficking could explain why STING’s canonical function and signaling is compromised during EV-D68 infection, as observed in Figure 3.

### 3.5. STING Modulates EV-D68-Induced Immunometabolism

It has recently emerged that respiratory viruses orchestrate host immunity by reprogramming immune cell metabolism [31]. In order to determine whether EV-D68 also modulates the immunometabolism of host immune cells, a bioinformatics analysis of the bulk RNA-sequencing profiling of bronchial epithelial cells infected with EV-D68 over multiple timepoints [23] was performed. The total RNA from non-infected and EV-D68-infected cells was analysed using RNA-sequencing to quantify differentially expressed genes (DEGs) relative to the mock control group. DEGs at 12 hpi (Figure 5A,B) and 24 hpi (Figure 5C) showed the upregulation of genes involved in the glycolysis pathway. Out of the thirteen DEGS at 12 phi (ALDOA, ALDOC, ENO1, ENO2, GPI, HK1, PFKB3, PFKP, PGKI, PKM, SLC2A1, HK2, and PFKB4) and thirteen DEGs at 24 hpi (ALDOA, ENO1, ENO2, GCK, GPI, HK1, HK2, HKDC1, PFKB3, PFKB4, PFKL, PFKP, and SLC2A1), ten DEGs (ALDOA, ENO1, ENO2, GPI, zHK1, HK2, PFKB3, PFKFB4, PFKP, and SLC2A1) were upregulated at both time points. In particular, glucose transporter 1 (GLUT1), also known as solute carrier family 2, facilitated glucose transporter member 1 (SLC2A1), encoded by the *SLC2A1* gene, which was highly upregulated. The results indicate, for the first time, a robust glycolysis-related gene signature in response to EV-D68 infection.

Since transcriptomics suggested the involvement of the glycolysis pathway, we proceeded to investigate the glucose metabolism in EV-D68-infected ALI cells. We utilised FRET in order to determine whether STING co-localises with glycolytic enzymes in ROs following EV-D68 infection. It was shown that, upon EV-D68 infection, STING co-localised with PI4P as well as glycolytic enzymes, such as HK, PK, and ALDOA, as well as SLC2A1 (GLUT1) in ROs (Figure 5D).

Our transcriptomic and FRET data suggested that EV-D68 manipulates the host metabolic programming, utilising glucose to fuel its replication. This glucose uptake increased upon EV-D68 infection (Figure 5E), as well as HRV (Figure 5F), SARS-CoV-2 (Figure 5G), and IAV infection (Figure 5H), suggesting that it must be a common mechanism among RNA viruses that induce ROs. Interestingly, this increase in glucose uptake could be inhibited when we treated the cells with a STING antagonist (Figure 5E–G), suggesting that STING regulates the metabolic reprogramming in infected cells from within Ros, therefore, suggesting that ROs are not just compartments utilised for viral replication but are also sites of a “glycolytic metabolon” for an energy supply to successfully achieve virion replication and assembly.

## 4. Discussion

The recent global pandemic has demonstrated that new, respiratory, RNA viral infections will always emerge in the absence of drugs for these infections. Even before the COVID-19 pandemic, viral respiratory tract infections accounted for more deaths than HIV, tuberculosis, and malaria combined. Following the pandemic, there are now fears that we might be faced by a tripledemic, with SARS-CoV-2 joined with Influenza and RSV.

The innate immune system, especially Type I IFNs, play a major role in triggering the initial antiviral response against these viruses. STING has been shown to have many functions including the sensing of intracellular DNA via the cGAS-STING pathway, resulting in triggering an interferon (IFN) and inflammatory response [9,10,11,12], as well as inducing autophagy for the clearance of cytosolic DNA [13]. In contrast to its well-documented role in antiviral immune responses to DNA viruses, the STING function in RNA virus infections is largely unexplored.

Studies have shown that cGAS-STING can be activated indirectly by mislocalised nuclear DNA or mtDNA released from the damage imposed by the RNA virus [32] leading to host antiviral immune responses [33]. However, this has not been shown for respiratory RNA viruses. Our recent work revealed a novel non-canonical role for STING in RNA viral replication [8], where STING is recruited in ROs and is actually essential for HRV replication [8].

Our recent work demonstrated that, during HRV infection, STING was shown not to be involved in host antiviral responses. On the contrary, STING was essential for viral replication since it interacted with Phosphatidylinositol 4-phosphate (PI4P) and was trafficked to replication organelles (ROs) created by the virus to facilitate HRV replication and transmission via autophagy [8].

The question now is whether this is a general mechanism across RNA viruses—are STING and PI4P also important for other respiratory RNA viruses, such as Enterovirus D-68 (EV-D68), or coronaviruses (HCoVs), which also generate Ros?

A bioinformatics analysis of single-cell and bulk RNA-seq publicly available databases from EV-D68 as well as COVID-19 infections [23,24] revealed a significant upregulation of cGAS-STING signalling pathway genes (TMEM173, IRF3, NFKB1, CGAS, IFNAR1, and TBK1) as well as a profound transcriptional dysregulation of host genes of the PI4P/OSBP pathway (Figure 1A,B), thus suggesting the importance of STING as well as the PI4P pathway for RO-generating RNA viruses.

Since the bioinformatics analysis suggested the involvement of STING in EV-D68 infections, we proceeded to investigate its localisation in response to these viruses. Our confocal imaging data demonstrated that both EV-D68 as well as SARS-CoV-2 resided in ROs, and co-localised with STING (Figure 1C).

To validate the involvement of STING in EV-D68 infections, we proceeded to genetically and pharmacologically inhibit STING in ALI cultures and assess the viral replication. It was shown that EV-D68 was unable to replicate in STING-KO cells. Similarly, the pharmacological inhibition of STING using STING antagonists prior to EV-D68 infection also demonstrated the inhibition of viral replication, suggesting that STING is essential for EV-D68 replication. As a control, we tested HRV, which we have previously shown to have a STING-dependent replication, as well as SARS-CoV-2 and IAV. SARS-CoV-2, which forms ROs, was also shown to depend on STING for replication, whereas IVA, which does not form ROs, was not affected by either the genetic or pharmacological inhibition of STING (Figure 2), thus, suggesting that this might be a general mechanism for RNA viruses that utilise ROs.

However, different RNA viruses use different lipid-rich membranes for replication organelle biogenesis in addition to PI4P. For SARS-CoV-2, this was shown to be through a signaling lipid phosphatidic acid (PA) [34]. Therefore, it could very well be that the STING relocation to ROs of SARS-CoV-2 could play a role in enrichment of PA at SARS-CoV-2 ROs. This is something that we plan to explore in future studies.

Interestingly, when we investigated whether IFN production was impaired due to the STING relocation to ROs, it was shown that IFN production remained unaffected in the response to EV-D68 infection (Figure 3). STING did not seem to be involved in the host’s response to these viruses. In contrast, RIG-I/MDA5 was shown to be responsible for the innate immune sensing of these viruses and the subsequent IFN response. Similar to what we had previously observed for HRV, STING seems to function non-canonically in response to EV-D68. It is recruited in replication organelles and in order to support the replication of the virus and does not take part in the host’s response to infection by these viruses.

Interestingly, these ROs function not only as replication sites, but also as compartments where the energy needed for replication is concentrated. Our work demonstrated that EV-D68 co-localises with the glycolytic machinery of the cell within ROs (Figure 5); most likely, it hijacks the glycolytic enzymes in order to supply the constant energy needed for its replication, although this we will investigate further in future studies. The inhibition of STING led to the modulation of glucose metabolism (Figure 5), either by directly affecting immunometabolism or reducing glucose uptake via reduced viral replication which promotes glycolytic metabolism. Therefore, in EV-D68 infections, STING is important for its non-canonical functions in membrane lipid re-modelling, and the formation of replication vesicles to support viral replication, as well as immunometabolism—essential for fueling the replication. This non-canonical function of STING is not only restricted to EV-D68 but seemed to be common to both RO-forming RNA viruses (HRV and SARS-CoV-2) that we tested. This has recently been shown in bacterial infections, where STING has been found to regulate metabolic reprogramming in macrophages during *Brucella* infection [35]. Questions that remain include how STING causes this glycolytic metabolic shift in respiratory RNA viral infections.

It seems that EV-D68, like HRV, has evolved to exploit STING and use it as an integral component of its replication machinery, both as a component of the ROs but also as a regulator of the metabolic response that fuels the viral replication.

Our data suggest that targeting STING could be a novel preventive and effective treatment strategy to limit viral replication. This could be particularly beneficial due to the rapid emergence of new strains of respiratory RNA viruses such as EV-D68 and COVID, which evade naturally acquired or vaccine-acquired immunity, whereas targeting STING or PI4KB should provide a high barrier to resistance due to its mode of action.

## 5. Conclusions

Multiple viruses that utilise ROs have evolved to exploit STING as an integral component of their replication machinery.

## Figures and Tables

**Figure 1 viruses-16-01541-f001:**
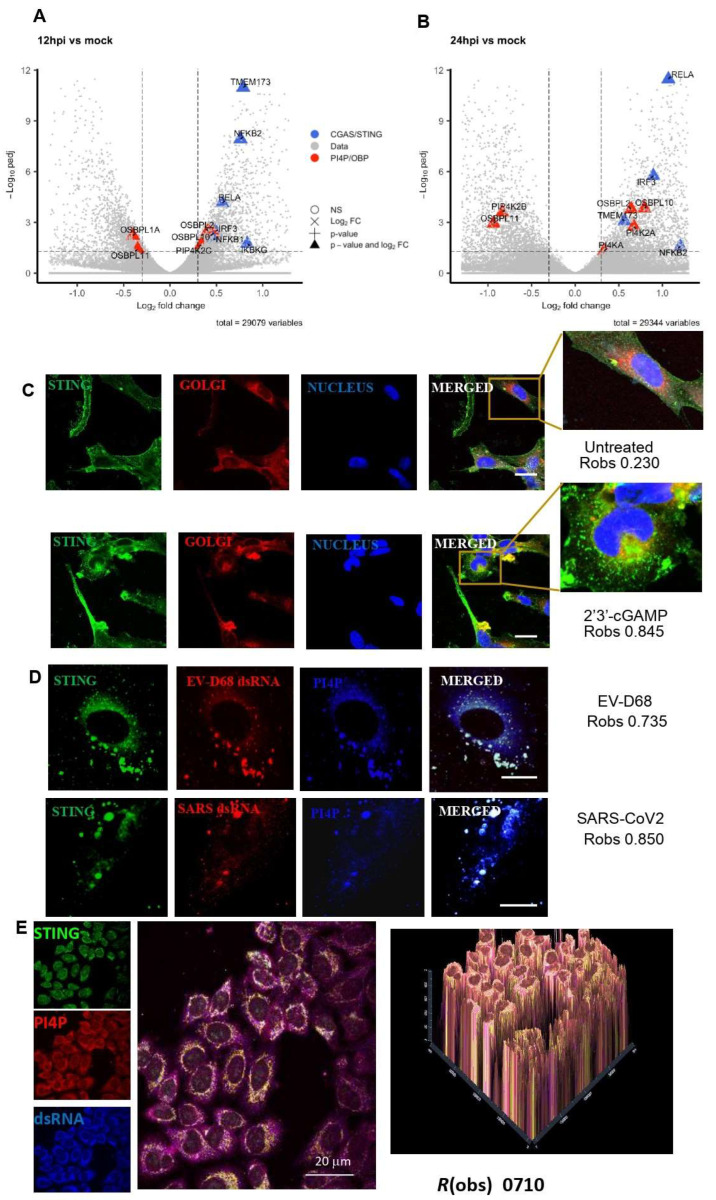
STING is involved in EV-D68 infection. Bioinformatics analysis of bulk RNA-sequencing profiling of cells infected with EV-D68 over multiple timepoints [23] (**A**,**B**). Volcano plots representing differential expression analysis results from [23], comparing DEGs in EV-D68-infected cells relative to mock control group at 12 hpi (**A**) and 24 hpi (**B**). Highlighted are the DEGs involved in cGAS/STING (blue) and PI4P/OSBP (red) pathways. At 12 hpi and at 24 hpi, 11 genes and 10 genes were differentially expressed, respectively, using the threshold values |log2FC| > 0.3 & padj < 0.05. (**C**) STING resides in the ER and does not co-localise with the Golgi in untreated cells ((**C**), top panel), whereas, upon stimulation with its ligand, 2′3′-cGAMP, it translocates to the Golgi apparatus ((**C**), bottom panels). (**D**) STING co-localises with EV-D68 (top panel) as well as SARS-CoV-2 (bottom panel) in bronchial epithelial cells as well as in ALIs (**E**). Bronchial epithelial cells (**D**) or ALIs (**E**) were infected with EV-D68 (**D**,**E**) or SARS-CoV-2 (**D**) (moi 10) for 4 h. Cells were fixed and permeabilised in PBS/0.02% BSA, prior to fixation with 4% formaldehyde for 15 min, then stained with J2 mAb to label the virus dsRNA, followed by the appropriate antibodies for the virus and PI4P. Cells were imaged using a Zeiss 710 confocal microscope. The degree of co-localisation, *R*(obs), was determined using ImageJ software as Pearson’s correlation coefficient (r) via the Costes’ method. Scale bar 5 mm is shown.

**Figure 2 viruses-16-01541-f002:**
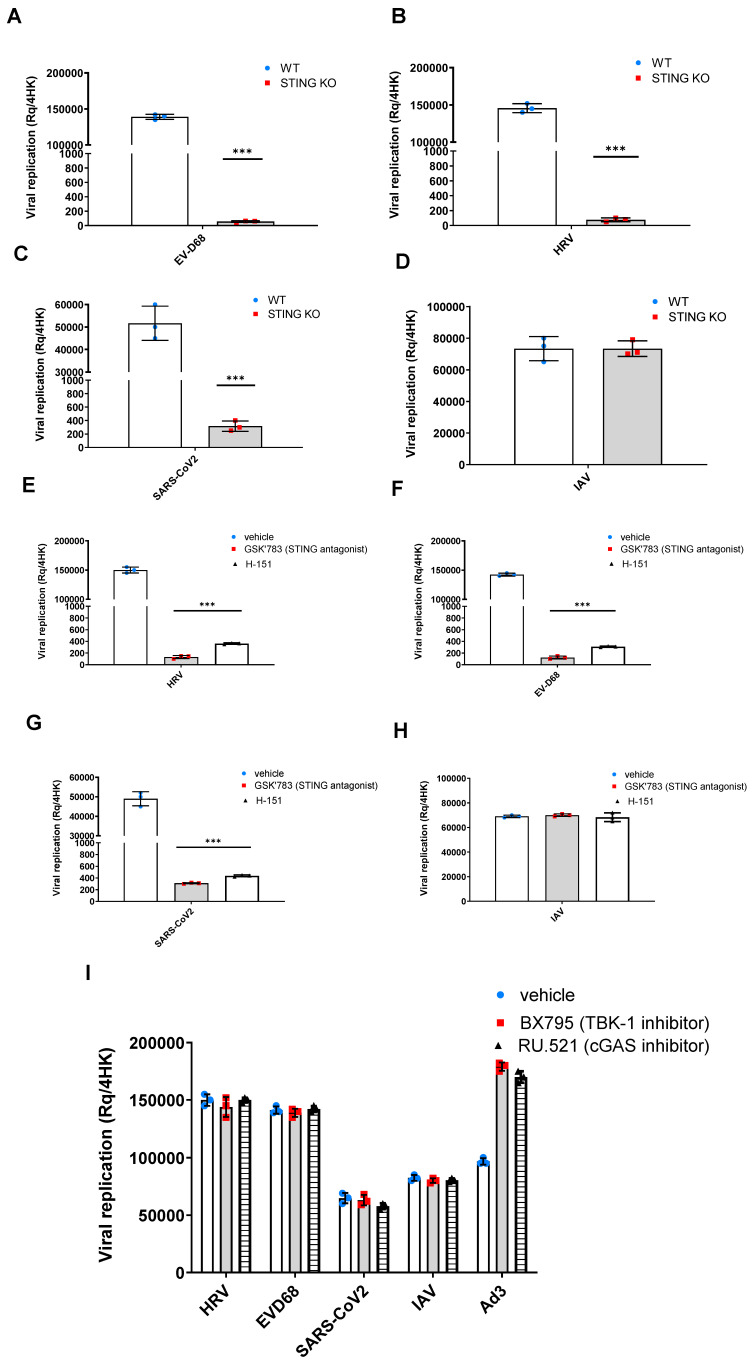
STING is required for EV-D68 replication. Airway epithelial (ALI)/STING KO clones were infected with EV-D68 (**A**) as well as other RNA respiratory viruses such as Human Rhinovirus (HRV) (**B**), SARS-CoV-2 (**C**), or Influenza A virus (IAV) (**D**) as controls. Viral replication was assessed using qPCR. Data are means ± SD (*n* = 3). Effectiveness of STING antagonists to inhibit Human Rhinovirus 1B (HRVA-1B), EV-D68, and SARS-CoV-2 viral replication in air–liquid interface (ALI) cultures (**E**–**H**). ALI cultures were either pre-treated for 1 h prior to infection with EV-D68 (**E**), HRVA-1B (**F**), SARS-CoV-2 (**G**), or IAV (H3N2) (**H**) with 0.1 µM STING antagonist GSK′783 or H-151; or either TBK-1 inhibitor BX795 or cGAS inhibitor RU.521 (**I**). Viral replication was assessed by qPCR 24 h after viral infection with MOI 1, demonstrating that STING antagonists were able to inhibit only RNA viruses that require ROs for replication. Data are represented as mean ± SD from three independent experiments. ***, *p* < 0.001.

**Figure 3 viruses-16-01541-f003:**
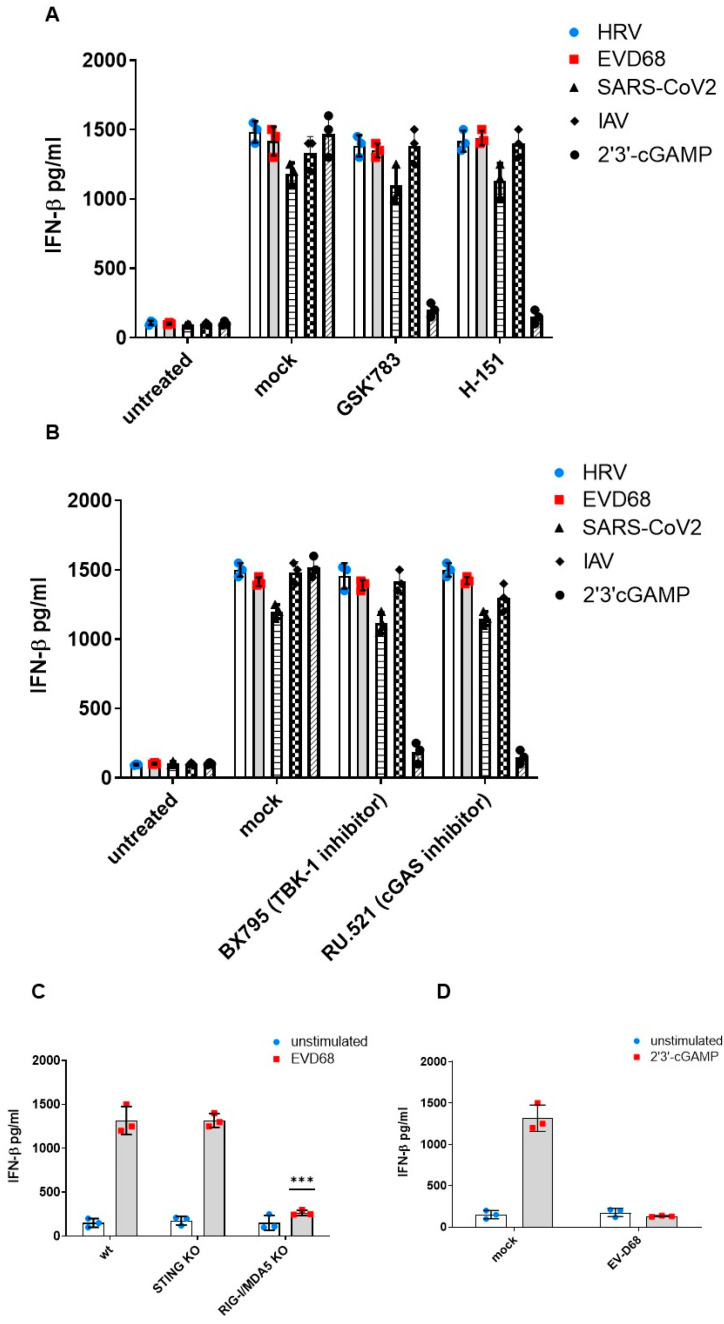
STING does not contribute to the Type-I interferon production in response to EV-D68. IFN-β production in BEAS-2B cells: (**A**) infected with EV-D68 in the presence of STING antagonists GSK 783 (0.1 µM) or H-151 for 24 h measured by ELISA, and (**B**) in the presence of inhibitors BX795 (1µM) and RU.521 (500 nM) for 24 h measured by ELISA (**C**). IFN-β production in BEAS-2B cells KO for STING or RIG-I/MDA5 and infected with EV-D68 (**C**). IFN-response to 2′3′-cGAMP following EV-D68 infection (**D**). The data represent the mean of three independent experiments ± SD (*n* = 3) yielding consistent results. ***, *p* < 0.001.

**Figure 4 viruses-16-01541-f004:**
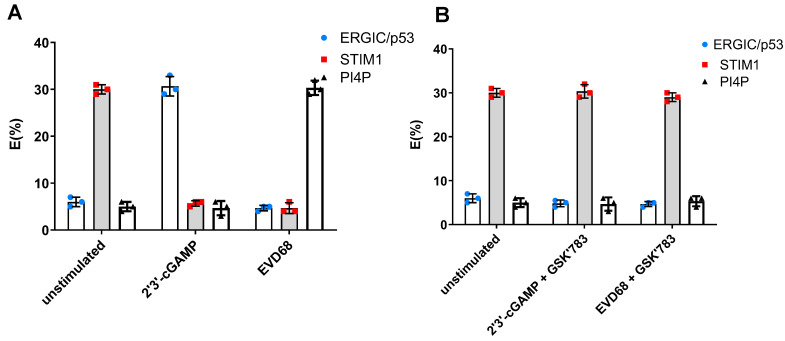
STING is trafficked to PI4P rich replication organelles in the presence of EV-D68. FRET studies measuring donor (STING) and acceptor (ERGIC, STIM1, or PI4P) interactions in BEAS-2B cells infected with EV-D68 for 2 h or exposed to 2′3′-cGAMP (1 µg) in the absence (**A**) or presence of STING antagonist GSK′783 (**B**). Data are means +/− SD (*n* = 3).

**Figure 5 viruses-16-01541-f005:**
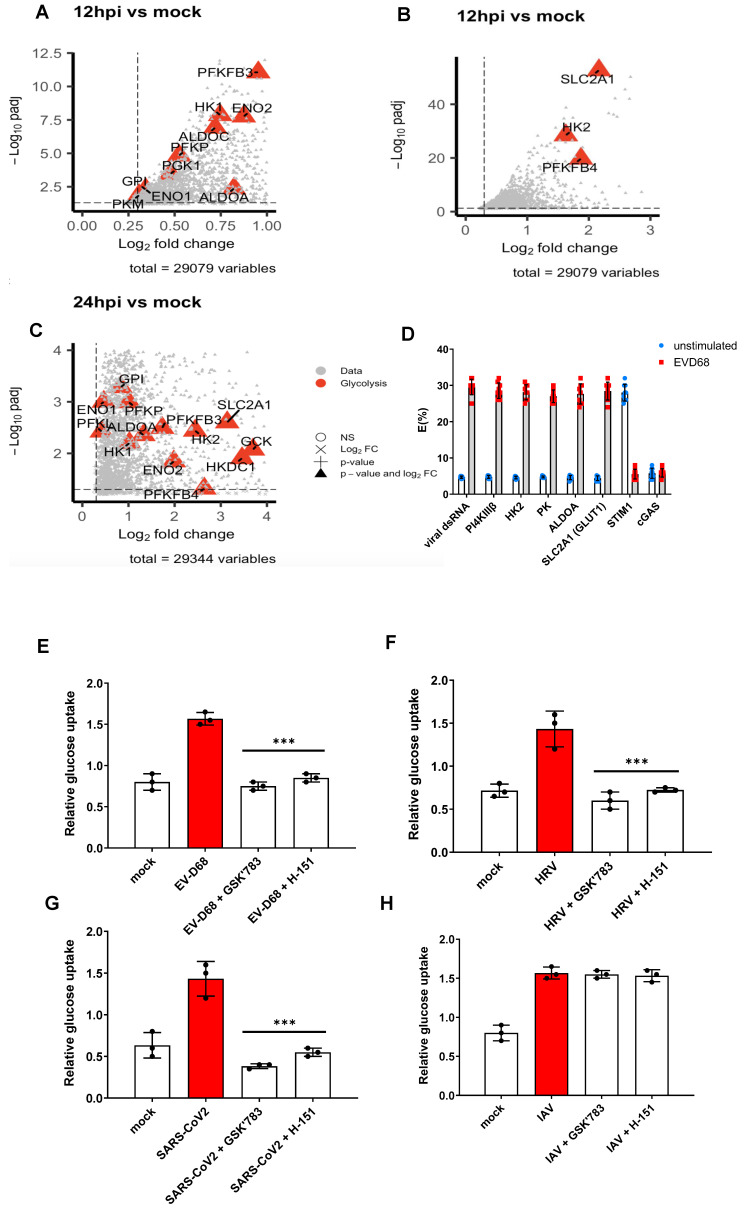
STING modulates EV-D68-induced glucose uptake. Bioinformatics analysis of bulk RNA-sequencing profiling of cells infected with EV-D68 over multiple timepoints [23] (**A**–**C**). Highlighted are key DEGs involved in the glycolysis pathway, using the threshold values |log2FC| > 0.3 & padj < 0.05. (**A**,**B**) shows upregulated DEGs for 0.3 < log2FC < 1 and for log2FC > 1 at 12 hpi, and (**C**) corresponds to positive DEGs at 24 hpi. (**D**) FRET studies measuring donor (STING) and acceptor (viral dsRNA, PI4KIIIβ, HK2, PK, ALDOA, SLC2A1, STIM1, and cGAS) interactions in BEAS-2B cells infected with EV-D68 for 2 h. Data are means +/− SD (*n* = 3). (**E**–**H**) Representative measurement of the uptake of fluorescently labeled glucose (2-NBDG) in ALI cells that were EV-D68 (**E**), HRVA-1B (**F**), SARS-CoV-2 (**G**), or IAV (**H**)-infected (MOI 1),-mock-infected, or -pre-treated 1 h prior to infection with 0.1 µM STING antagonist (GSK’783) or H-151; 6 h post infection is shown. Mean ± SD of three independent experiments. ***, *p* < 0.001.

## Data Availability

All the relevant data are available from the authors. The source data are provided with this manuscript.

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
