# Peer review of "STING Orchestrates EV-D68 Replication and Immunometabolism within Viral-Induced Replication Organelles"

_viruses, 2024, doi:10.3390/v16101541_

Round 1

Reviewer 1 Report

Comments and Suggestions for Authors

In the present manuscript, Triantafilou et al. have shown the proviral role of STING for the replication of EV-D68, HRV, and SARS-CoV-2 viruses. The authors showed that Type-I IFN signaling in EV-D68 does not occur via cGAS-STING pathway and is primarily via MDA5/RIG-I yet STING contributes to the replication cycle. Though STING gets recruited to the replication organelle sites, it is not clear what the role of STING is there. They suggest that it could be by up regulating the enzymes in glycolysis and PI4P lipid recruitment to the viral replication organelles (ROs) and that this is general mechanism for RNA viruses that generate ROs. The finding that STING plays a role in replication of these viruses is robust, however, the generalisation of STING role at the viral ROs by enriching PI4P and glycolytic enzymes is premature. Also, there are some concerns of how the data is presented which makes it difficult to judge experiments at some places. These points are described bellow in detail;

  • Figure 3A and 3B - There are no no-infection or no-treatment datapoints. Difficult to judge the increase due to these inducers, especially for 2’3’ cGAMP and STING inhibitors. Whether STING signalling is functional in these cells or if IFN signalling is in general independent of cGAS-STING pathway in this system? This is also true for Fig 2I which can benefit from a positive control, eg polyIC, herpes simplex, or adenovirus.
  • The authors should harmonise their findings in terms of STING signalling. The initial point of the study was that STING dependent DEGs were up-regulated in infection. Fig 3D suggests that EV-D68 actually inhibits canonical STING signalling. Does non-canonical STING signalling allows only a subset of DEGs through canonical STING signalling? This could be checked by measuring these DEGs in actual EV-D68 infected cells treated with STING inhibitors using conventional qPCR based assays (using samples in Fig 2E). Since the data suggests that EV-D68 induces Type I IFN signalling via MDA5/RIG-I, hence, MDA5/RIG-I activation could be the inducers of these DEGs during EV-D68 infection.
  • It is clear STING relocates to PI4P membranes in EV-D68 infection. However, different RNA viruses use different lipid rich membranes for replication organelle biogenesis. For SARS-CoV-2, this was shown to be through a signaling lipid phosphatidic acid (PA)(PMID: 34907161). Hence, generalisation of PI4P requirement for EV-D68 ROs observation for other viruses (eg SARS-CoV-2) without experimentally testing the data is a bit of a stretch and premature. It could very well be that STING relocalisation to ROs of SARS-CoV-2 could play a role in enrichment of PA at SARS-CoV-2 ROs. Hence, I would suggest this to be discussed as an outlook fur future studies.
  • The FRET data only shows that STING colocalises with glycolytic enzymes at the ROs. It does not suggest that it actively recruits these enzymes to the ROs. Also, STING inhibitors reducing glucose uptake could simply be via reduced viral replication which promotes glycolytic metabolism. Hence, the role of STING in RO glycolytic immunometabolism is only correlative and may not be causal in nature. The authors should clearly state this.

Overall, I think the manuscript is a nice work but can be improved by not generalising principles without experimental proof.

Comments on the Quality of English Language

The quality of the English is sufficient for understanding at most places. However, some typos and grammatical improvements could be done. 

Author Response

Figure 3A and 3B - There are no no-infection or no-treatment datapoints. Difficult to judge the increase due to these inducers, especially for 2’3’ cGAMP and STING inhibitors. Whether STING signalling is functional in these cells or if IFN signalling is in general independent of cGAS-STING pathway in this system? This is also true for Fig 2I which can benefit from a positive control, eg polyIC, herpes simplex, or adenovirus.

We agree with the reviewer. We have added no treatment datapoints for all graphs. For Figure 2I we have added a positive control with adenovirus infection.

The authors should harmonise their findings in terms of STING signalling. The initial point of the study was that STING dependent DEGs were up-regulated in infection. Fig 3D suggests that EV-D68 actually inhibits canonical STING signalling. Does non-canonical STING signalling allows only a subset of DEGs through canonical STING signalling? This could be checked by measuring these DEGs in actual EV-D68 infected cells treated with STING inhibitors using conventional qPCR based assays (using samples in Fig 2E). Since the data suggests that EV-D68 induces Type I IFN signalling via MDA5/RIG-I, hence, MDA5/RIG-I activation could be the inducers of these DEGs during EV-D68 infection.

We agree with the reviewer that it would have been a nice addition to measure DEGs in actual EV-D68 infected cells treated with STING inhibitors, but unfortunately we did not keep the samples used in Fig. 2E and thus it would have taken too long to perform the experiments again and also perform qPCR on the lysates produced – which would have been outside of the time given for completing these corrections. Although we feel it would have been a nice addition to the manuscript, it is not essential for the conclusions drawn from this study.

It is clear STING relocates to PI4P membranes in EV-D68 infection. However, different RNA viruses use different lipid rich membranes for replication organelle biogenesis. For SARS-CoV-2, this was shown to be through a signaling lipid phosphatidic acid (PA)(PMID: 34907161). Hence, generalisation of PI4P requirement for EV-D68 ROs observation for other viruses (eg SARS-CoV-2) without experimentally testing the data is a bit of a stretch and premature. It could very well be that STING relocalisation to ROs of SARS-CoV-2 could play a role in enrichment of PA at SARS-CoV-2 ROs. Hence, I would suggest this to be discussed as an outlook for future studies.

We agree with the reviewer and have added this statement in our discussion.

The FRET data only shows that STING colocalises with glycolytic enzymes at the ROs. It does not suggest that it actively recruits these enzymes to the ROs. Also, STING inhibitors reducing glucose uptake could simply be via reduced viral replication which promotes glycolytic metabolism. Hence, the role of STING in RO glycolytic immunometabolism is only correlative and may not be causal in nature. The authors should clearly state this.

We agree with the reviewer and have re-worded our statements to reflect this in our discussions as well as in the abstract of the revised manuscript. 

Reviewer 2 Report

Comments and Suggestions for Authors

Some ssRNA(+) viruses, including EV-D68, SARS-CoV-2 and HRV, produce organelle-like structures in the ER called replication organelles (ROs), which are the site of viral genome replication.

This paper is a follow-up to another paper in Nature Communications (https://doi.org/10.1038/s41467-022-28745-3), in which the same group published observations that HRV replication and transmission requires STING interaction with PI4P in the ROs. In the study submitted to Viruses, the authors investigate whether STING also plays a similar role in the replication of EV-D68, SARS-CoV-2 and an influenza A virus. As expected, the influenza A virus does not seem to require STING as it replicates in the nucleus. Interestingly, replication of EV-D68 and SARS-CoV-2 was abrogated by knockdown or inhibition of STING. Experiments also showed the co-localization of STING with viral RNA in ROs, the role of STING in IFN-I production, and the role of STING in glucose uptake during viral infection.

Overall, the experiments performed in the paper were able to show the non-canonical role of STING in RNA viruses that induce the formation of ROs. The paper is well written and guides the reader to understand the results and arrive at the author's conclusions.

The authors conclude that the previously published STING function in ROs formation and HRV replication also applies to other ssRNA(+) such as EV-D68 and SARS-CoV-2. Therefore, I would suggest to change the title of the article to include SARS-CoV-2.

Minor corrections:

1)        The usual acronym for Influenza A viruses is IAVs and I would suggest is in this article to avoid confusion with the interferon acronym (IFN) as it happened in the discussion (line 403).

2)        The IAV strain used in the work is not described in the “cell culture/viruses” part of the methods and it is not described in the paper.

3)        Also, in the “cell culture/viruses” part: which cells were used for ALI culture? Human Nasal Epithelial Cells (hNECs) or Human Bronchial Epithelial Cells (hBECs)? Or were the ALIs derived from the BEAS-2B cells? This information should be clear in the main text as well in the Figures.

4)        In the methods section, the last paragraph of “Quantification of co-localization” is misplaced there. The “CRISPR STING knockout” part is messed up; it has the same phrases repeated in a row.

5)        Figure 1C. Scale is missing. Please show a mock uninfected control to visualize the distribution of STING and PI4P.

6)        Line 259. The Figure 2A does not show the production of IFN-betta by ALI cells in response to 2’3’-cGAMP nor to EV-D68 infection.

7)        Figure 3 shows the results of experiments performed in BEAS-2B cells; however, the section 3.3 describes that the results were obtained by infection experiments in ALI cells.

8)        Line 301/302: these results are from Figure 4A and not Figure 3A.

9)        Line 304: the text says that inhibition of STING during EV-D68 makes STING remain in the ER associated with STIM1. This is shown in Figure 4B, and the STING antagonist used is GSK’783. However, the inhibitor H-151 is not shown as written in Line 304.

10)   Line 308. Figure 2 is referred wrong. Should it be Figure 3 instead?

11)   Figure 5H. Influenza A virus infection, like EV-D68, HRV, and SARS-CoV-2 infection in BEAS-2B cells, also seems to have increased the glucose uptake. This is not what is stated in Line 340. It would be relevant to show the statistic significance of the increased glucose uptake in comparison to the uninfected mock.

Author Response

1)        The usual acronym for Influenza A viruses is IAVs and I would suggest is in this article to avoid confusion with the interferon acronym (IFN) as it happened in the discussion (line 403).

We agree with the reviewer and have changed the acronym for Influenza A viruses to IAV throughout the revised manuscript.

2)        The IAV strain used in the work is not described in the “cell culture/viruses” part of the methods and it is not described in the paper.

We agree with the reviewer and have added information in the “cell culture/viruses” section of the methods regarding the IAV strain used.

3)        Also, in the “cell culture/viruses” part: which cells were used for ALI culture? Human Nasal Epithelial Cells (hNECs) or Human Bronchial Epithelial Cells (hBECs)? Or were the ALIs derived from the BEAS-2B cells? This information should be clear in the main text as well in the Figures.

We have used Human Bronchial Epithelial cells (hBECs) and we have added this information in the methods section as well as in the figures of the revised manuscript.

4)        In the methods section, the last paragraph of “Quantification of co-localization” is misplaced there. The “CRISPR STING knockout” part is messed up; it has the same phrases repeated in a row.

We agree with the reviewer and we have amended the “Quantification of co-localisation” as well as the “CRISPR STING knockout” sections in the methods.

5)        Figure 1C. Scale is missing. Please show a mock uninfected control to visualize the distribution of STING and PI4P.

We agree with the reviewer and have added the missing scale bar in Figure 1C. In addition, we have added a mock uninfected control as well as a positive control with cells stimulated with cGAMP in the revised Figure 1 of the manuscript.

6)        Line 259. The Figure 2A does not show the production of IFN-betta by ALI cells in response to 2’3’-cGAMP nor to EV-D68 infection.

We agree with the reviewer and we have amended the text describing Figure 2A.

7)        Figure 3 shows the results of experiments performed in BEAS-2B cells; however, the section 3.3 describes that the results were obtained by infection experiments in ALI cells.

We agree with the reviewer and have amended the text in the section 3.3 which describes Figure 3.

8)        Line 301/302: these results are from Figure 4A and not Figure 3A.

We agree with the reviewer and have amended the text in order to refer to Figure 4A.

9)        Line 304: the text says that inhibition of STING during EV-D68 makes STING remain in the ER associated with STIM1. This is shown in Figure 4B, and the STING antagonist used is GSK’783. However, the inhibitor H-151 is not shown as written in Line 304.

We agree with the reviewer and have amended the text in order to remove the reference to inhibitor H-151.

10)   Line 308. Figure 2 is referred wrong. Should it be Figure 3 instead?

We agree with the reviewer and we have amended the text to refer to Figure 3 instead of Figure 2.

11)   Figure 5H. Influenza A virus infection, like EV-D68, HRV, and SARS-CoV-2 infection in BEAS-2B cells, also seems to have increased the glucose uptake. This is not what is stated in Line 340. It would be relevant to show the statistic significance of the increased glucose uptake in comparison to the uninfected mock.

We agree with the reviewer, Figure 5H shows that IAV also seems to have increased glucose uptake. Therefore, we have amended the text in order to reflect this.

Reviewer 3 Report

Comments and Suggestions for Authors

In their paper “STING orchestrates EV-D68 replication and immunometabolism within viral-induced replication organelles”, Triantafilou et al. describe how STING is necessary for EV-D68 and SARS-CoV-2 replication. In the absence of STING, these viruses appear to be unable to form replication organelles. The findings are novel and interesting, but a number of controls are lacking, and part of the conclusion is not supported by the data.

Major issues:

I miss a couple of controls in figure 1C and 1D. First, one control without infection with a virus to show localization of STING in the absence of infection and second, a control with cGAMP to show the localization of activated STING.

---

In figure 2, there is no verification that the knockout cells are actually knockout for STING! A western blot and/or a functional assay (treating them with cGAMP or diABZI and see that IFN production is gone) and comparing knockout and wild-type cells is a must!

Likewise, verification that the STING, cGAS and TBK1 antagonists work as intended and inhibit STING signaling in that specific experimental setup is also a must!

---

Where are the untreated cells, i.e. cells that are not infected or treated with cGAMP, in figure 3A and 3B? They need to be included.

---

In figure 3C, there is no verification that the knockout cells are actually knockout for STING or RIG-I/MDA5! A western blot and/or a functional assay using an agonist and comparing knockout and wild-type cells is a must!

---

In general, I lack information about the STING antagonist GSK’783 that the authors use in their study. I couldn’t find any information in Materials and Methods, and I also couldn’t find it when I tried Google. Is the name misspelled? Is it known how it inhibits the function of STING?

---

In figure 4A and 4B, inactive STING is shown to interact with STIM1 but in figure 5D, inactive STING suddenly no longer interacts with STIM1. How can this be the case? This raises questions about the validity of the presented data.

---

The authors write that “This glucose uptake increased upon EV-D68 infection (Fig 5E), as well as HRV (Fig 5F) and SARS-CoV2 (Fig 5G) infection but not in response to IFVA (Fig 5H)” (line 338-340). However, figure 5H does show increased glucose uptake upon IAV infection so their own data does in fact contradict their statement.

---

The authors write that “STING regulates the metabolic reprogramming in infected cells from within ROs.” (line 342-343). However, there is not enough data to support that conclusion! What the authors can conclude is that 1) EV-D68 infection causes metabolic reprogramming and 2) STING is necessary for EV-D68 replication. They cannot conclude that it is STING that regulates the metabolic reprogramming! The abstract and the conclusion must be modified accordingly to make it crystal clear what can actually be concluded based on the presented data.

---

I lack information about figure S1 and how it was made. Is this just comparison between lung and other organs from COVID-19 patients? The way it is presented in the main text, I thought the authors were trying to claim that SARS-CoV-2 infection upregulates cGAS-STING signaling pathway genes in the lung. However, based on the figure legend it seems that the data is from COVID-19 patients only. In that case, the figure only shows that cGAS-STING signaling pathway genes are in general more highly expressed in the lung than in many other tissues, which is already known.

Minor issues:

There is an issue with the text from line 141 to 149 where the same sentence is repeated again and again.

---

The figure legend for figure 1C is unclear. Which virus where the cells infected with?

---

The abbreviation for influenza A virus is IAV (or less commonly used FLUAV) and not IFVA.

---

Line 12: The main anti-viral sensing system of the innate immune system controlling the type I IFN machinery…

Line 40-42: The cyclic GMP-AMP synthase (cGAS) - stimulator of interferon genes (STING) pathway is the main anti-viral sensing system of the innate immune system controlling the type I IFN machinery…

Line 253-255: with the cGAS-STING pathway as the main anti-viral DNA sensing system controlling the type I IFN machinery…

I think a lot of people working on RLRs or TLRs would disagree with that! I suggest you change it to “one of the main antiviral sensing systems…”

---

Line 14: Although it is well characterized as a DNA sensor…

STING is not a DNA sensor! The DNA is sensed by cGAS! STING senses the cGAMP made by cGAS.

---

Line 45-46: Our work has revealed a novel non-canonical role for STING in RNA viral infections…

At this point, you have not yet established that this is true for RNA viruses in general so RNA viruses should be replaced with HRV

---

The authors are inconsistent in the way they introduce abbreviations in the abstract. Please state the full name of SARS-CoV2 and IFN.

---

Line 55-57: Together with coronaviruses, HRVs are responsible for the majority of respiratory tract infections in all age groups…

Do you mean historically or presently?

---   

Line 258-259: It was shown that ALI cells produced 258 IFN-β in response to 2’3’-cGAMP, but also EV-D68 infection (Figure 2A)

I believe the sentence should refer to figure 3A and not figure 2A.

---

Line 300-302: The data confirmed that in response to 2’3’-cGAMP, STING interacted with ERGIC and was not interacting with either STIM1 or PI4P (Fig 3A).

I believe the sentence should refer to figure 4A and not figure 3A.

---

Line 303-305: Pharmacological inhibition of STING during EV-D68 infection, using either GSK’783 or H-151, demonstrated that STING did not associate with PI4P but instead remained in the ER and associated with STIM1 (Fig. 4B)

There is no experiment with H-151 in that figure!

Author Response

I miss a couple of controls in figure 1C and 1D. First, one control without infection with a virus to show localization of STING in the absence of infection and second, a control with cGAMP to show the localization of activated STING.

We agree with the reviewer and have added a no treatment (no infection) control as well as a control stimulated with cGAMP.

In figure 2, there is no verification that the knockout cells are actually knockout for STING! A western blot and/or a functional assay (treating them with cGAMP or diABZI and see that IFN production is gone) and comparing knockout and wild-type cells is a must!

We agree with the reviewer and have added verification of STING knockout in the supplemental figure S1. We have verified knockout by flow cytometry (S1A) as well as using a functional assay (S1B) as the reviewer has suggested.

Likewise, verification that the STING, cGAS and TBK1 antagonists work as intended and inhibit STING signaling in that specific experimental setup is also a must!

We agree with the reviewer and have added verification of the function of the inhibitors in supplemental figure S2.

Where are the untreated cells, i.e. cells that are not infected or treated with cGAMP, in figure 3A and 3B? They need to be included.

We agree with the reviewer and have added untreated controls in all of the graphs where we have used stimulations.

In figure 3C, there is no verification that the knockout cells are actually knockout for STING or RIG-I/MDA5! A western blot and/or a functional assay using an agonist and comparing knockout and wild-type cells is a must!

We agree with the reviewer and have added verification of the RIG-I/MDA-5 knockout in supplemental figure S3. We have verified knockout by flow cytometry (S3A) as well as using a functional assay (S3B) as the reviewer has suggested.

In general, I lack information about the STING antagonist GSK’783 that the authors use in their study. I couldn’t find any information in Materials and Methods, and I also couldn’t find it when I tried Google. Is the name misspelled? Is it known how it inhibits the function of STING?

STING antagonist GSK’783 has been provided by Dr Joshi Ramanjulu from GSK. We have added this information in the methods section of the revised manuscript.

In figure 4A and 4B, inactive STING is shown to interact with STIM1 but in figure 5D, inactive STING suddenly no longer interacts with STIM1.

We agree with the reviewer and have repeated the experiment. In the revised figure, inactive STING interacts with STIM1.

The authors write that “This glucose uptake increased upon EV-D68 infection (Fig 5E), as well as HRV (Fig 5F) and SARS-CoV2 (Fig 5G) infection but not in response to IFVA (Fig 5H)” (line 338-340). However, figure 5H does show increased glucose uptake upon IAV infection so their own data does in fact contradict their statement.

We agree with the reviewer and we have amended the text in the results in order to accurately describe the figure.

The authors write that “STING regulates the metabolic reprogramming in infected cells from within ROs.” (line 342-343). However, there is not enough data to support that conclusion! What the authors can conclude is that 1) EV-D68 infection causes metabolic reprogramming and 2) STING is necessary for EV-D68 replication. They cannot conclude that it is STING that regulates the metabolic reprogramming! The abstract and the conclusion must be modified accordingly to make it crystal clear what can actually be concluded based on the presented data.

We agree with the reviewer and have amended the text in discussion as well as the abstract of the revised manuscript in order to reflect this.

I lack information about figure S1 and how it was made. Is this just comparison between lung and other organs from COVID-19 patients? The way it is presented in the main text, I thought the authors were trying to claim that SARS-CoV-2 infection upregulates cGAS-STING signaling pathway genes in the lung. However, based on the figure legend it seems that the data is from COVID-19 patients only. In that case, the figure only shows that cGAS-STING signaling pathway genes are in general more highly expressed in the lung than in many other tissues, which is already known.

Figure S4 (formerly S1) is data from COVID19 patients and it shows cGAS-STING pathway genes are generally more highly expressed in the lung than in other tissues. We have made this obvious in the revised manuscript.

There is an issue with the text from line 141 to 149 where the same sentence is repeated again and again.

We agree with the reviewer and have removed the repeated sentence.

The figure legend for figure 1C is unclear. Which virus where the cells infected with?

We have amended the figure legend in order to state which virus is being used.

The abbreviation for influenza A virus is IAV (or less commonly used FLUAV) and not IFVA.

We agree with the reviewer and have changed the abbreviation for Influenza A virus to IAV throughout the revised manuscript.

Line 12: The main anti-viral sensing system of the innate immune system controlling the type I IFN machinery…

We agree with the reviewer and have amended the text accordingly in the revised manuscript.

Line 40-42: The cyclic GMP-AMP synthase (cGAS) - stimulator of interferon genes (STING) pathway is the main anti-viral sensing system of the innate immune system controlling the type I IFN machinery…

We agree with the reviewer and have amended the text accordingly in the revised manuscript.

Line 253-255: with the cGAS-STING pathway as the main anti-viral DNA sensing system controlling the type I IFN machinery…I think a lot of people working on RLRs or TLRs would disagree with that! I suggest you change it to “one of the main antiviral sensing systems…”

We agree with the reviewer and have amended the text accordingly in the revised manuscript.

Line 14: Although it is well characterized as a DNA sensor…STING is not a DNA sensor! The DNA is sensed by cGAS! STING senses the cGAMP made by cGAS.

We agree with the reviewer and have amended the text accordingly in the revised manuscript.

Line 45-46: Our work has revealed a novel non-canonical role for STING in RNA viral infections…At this point, you have not yet established that this is true for RNA viruses in general so RNA viruses should be replaced with HRV

We agree with the reviewer and have amended the text accordingly in the revised manuscript.

The authors are inconsistent in the way they introduce abbreviations in the abstract. Please state the full name of SARS-CoV2 and IFN.

We agree with the reviewer and have amended the text accordingly in the revised manuscript.

Line 55-57: Together with coronaviruses, HRVs are responsible for the majority of respiratory tract infections in all age groups…Do you mean historically or presently?

We mean presently.

Line 258-259: It was shown that ALI cells produced 258 IFN-β in response to 2’3’-cGAMP, but also EV-D68 infection (Figure 2A). I believe the sentence should refer to figure 3A and not figure 2A.

We agree with the reviewer that it should be referring to Figure 3A. We have amended the text accordingly in the revised manuscript.

Line 300-302: The data confirmed that in response to 2’3’-cGAMP, STING interacted with ERGIC and was not interacting with either STIM1 or PI4P (Fig 3A). I believe the sentence should refer to figure 4A and not figure 3A.

We agree with the reviewer that it should be referring to Figure 4A. We have amended the text accordingly in the revised manuscript.

Line 303-305: Pharmacological inhibition of STING during EV-D68 infection, using either GSK’783 or H-151, demonstrated that STING did not associate with PI4P but instead remained in the ER and associated with STIM1 (Fig. 4B). There is no experiment with H-151 in that figure!

We agree with the reviewer and have amended the text accordingly in the revised manuscript.

Round 2

Reviewer 1 Report

Comments and Suggestions for Authors

The authors have addressed my comments. I recommend acceptance of this manuscript.

Comments on the Quality of English Language

Some sentences require phrasing in scientific communication.

Reviewer 3 Report

Comments and Suggestions for Authors

I believe the authors have addressed all the issues I raised